# Merchant Recommender System Using Credit Card Payment Data

Suyoun Yoo [1] and Jaekwang Kim [2,*]

1   Department of Applied Data Science, Sungkyunkwan University, Suwon 03063, Republic of Korea
2   School of Convergence/Convergence Program for Social Innovation, Sungkyunkwan University, Suwon 03063, Republic of Korea
*   Correspondence: linux@skku.edu

**Abstract:** As the size of the domestic credit card market is steadily growing, the marketing method for credit card companies to secure customers is also changing. The process of understanding individual preferences and payment patterns has become an essential element, and it has developed a sophisticated personalized marketing method to properly understand customers' interests and meet their needs. Based on this, a personalized system that recommends products or stores suitable for customers acts to attract customers more effectively. However, the existing research model implementing the General Framework using the neural network cannot reflect the major domain information of credit card payment data when applied directly to store recommendations. This study intends to propose a model specializing in the recommendation of member stores by reflecting the domain information of credit card payment data. The customers' gender and age information were added to the learning data. The industry category and region information of the settlement member stores were reconstructed to be learned together with interaction data. A personalized recommendation system was realized by combining historical card payment data with customer and member store information to recommend member stores that are highly likely to be used by customers in the future. This study's proposed model (NMF_CSI) showed a performance improvement of 3% based on HR@10 and 5% based on NDCG@10, compared to previous models. In addition, customer coverage was expanded so that the recommended model can be applied not only to customers actively using credit cards but also to customers with low usage data.

**Keywords:** merchant recommendation; credit card payment data; collaborative filtering; personalized recommender system

## 1. Introduction

The size of the credit card market is growing steadily. In 2019, the number of credit cards in Korea was 110.98 million, up 5.92 million from the previous year. In other words, each economically active population in Korea had 3.9 units [1]. In addition to policy support, such as income deductions, the use of credit cards has greatly expanded due to the increase in credit card preference for micropayment and simple payment services. Credit sales through credit cards accounted for only 13.7% of private final consumption expenditures in 2000 but have steadily risen since then, exceeding 70% [2]. When combined with debit cards, the utilization rate of non-cash electronic payment methods in Korea reaches 90% [3].

Accordingly, domestic credit card companies are changing their marketing methods to attract customers [4]. In the early days of card industry growth, marketing focused on card issuance and loans. Now, personalized marketing methods that provide optimized services for each customer are being developed extensively [5]. The mass method, which provides services to an unspecified number of people, is expensive, and the customer response rate is low. However, personalized marketing that properly understands customer interests and satisfies their needs can expect a high response rate at a relatively low cost.

Recently, in the credit card industry, the recommendation system has become an essential element for effective personalized marketing [6]. This is because understanding individual tastes and payment patterns and recommending suitable products or stores based on that data can effectively lock in customers. Therefore, this study proposes a personalized recommendation system model that recommends merchants that customers are likely to use in the future by combining historical credit card payment data with customer and member store information.

The model proposed in this study is based on [4], Neural Collaborative Filtering, which has been widely used in recent recommendation system research. The proposed model has three major changes.

First, unlike the existing model, which only reflects whether the user purchased the item, the customer's demographic information was added to the learning data. This is because customers of the same gender and age group were expected to show similar payment patterns [7].

Secondly, in order to reflect the unique characteristics of credit card payment data, information on the industry and region of merchants were added and learned together. The industry information is the most basic data representing the characteristics of the merchants, and regional information also acts as an important factor in understanding payment patterns. It is clear that recommending franchises in Busan to customers living in Seoul would reduce the possibility of future use [8].

Third, the model can be applied to customers with low card usage. In the existing model, only data with more than twenty interactions between users and items were used. However, by lowering the criteria to two, this study implemented a model that can be applied to customers with many interactions and customers with few interactions. Customers were divided into three groups (High/Mid/Low) according to the number of affiliated stores they used, and performance was compared for each group [9,10].

## 2. Related Works

### 2.1. Matrix Factorization

Matrix Factorization is a representative method of collaborative filtering. It is a technique of decomposing the interaction matrix between the user and item into a user-latent matrix and item-latent matrix through matrix decomposition [11].

The characteristics of the user and the item are captured through the Latent Vector, inferred from the Rating Data evaluated by the user, and the interaction between the user and the item is modeled through the dot product in the f-dimensional latent factor space. The vector for the user is $p_u$, and the vector for the item is $q_i$. Next, the predicted rating is calculated as follows.

$$\min_{p,q,b} \sum_{(u,i) \in K} c_{ui} \left( r_{ui} - \mu - b_i - b_u - q_i^T p_u \right)^2 + \lambda \left( \parallel q_i \parallel^2 + \parallel p_u \parallel^2 + b_u^2 + b_i^2 \right) \quad (1)$$

Figure 1 shows the structure of Matrix Factorization. As the Figure shows, MF uses user and item information. And it can be decomposed to X and Y matrices.

### 2.2. Explicit vs. Implicit Feedback Data

Explicit Feedback Data is data that directly expresses the user's preference for items [12]. For example, in the case of movie reviews, one can express their preference for the movie by rating it using 1 to 5 point and 1 to 10 point scales. Additionally, on Netflix, one can find the user's preferences directly through the like and dislike buttons. There is a strong advantage to determining the likes/dislikes of users, but there is a disadvantage in that it is difficult to collect data because users actively need to leave reviews and ratings after watching the movie.

On the other hand, Implicit Feedback Data refers to data that indirectly represents customers' preferences and tastes [13]. These include user search records, page visits, and online shopping purchase history. Although this data has the advantage of being relatively

easy to collect, it has the disadvantage in that it is difficult to accurately grasp preferences because there is no negative feedback. For example, when there is no record of a user purchasing item A, it is impossible to determine whether the user did not purchase the item because they did not prefer it or if they simply did not know about it. In this way, when dealing with Implicit Feedback Data, there is a possibility that the unobserved data may include the user's non-preferred information, so missing data should also be considered. In addition, there is a possibility that noise may be included in the data, such as the exact motive for purchasing item B and information on satisfaction after purchase. For example, it is not known whether the user who purchased item B may have preferred it or purchased it as a gift. In Explicit Feedback Data, if the number is high, it can be said that means a preference for the item, but in Implicit Feedback Data, preference is not always high. If one stayed on a page for a long time while using an online shopping mall, they might have liked the page, but they may also have left the page on for a while. Nevertheless, it can be interpreted that a high number in Implicit Feedback Data means reliable data. This is because it is more likely that videos that have been watched more frequently or for longer than videos that have been watched only once express the user's preference.

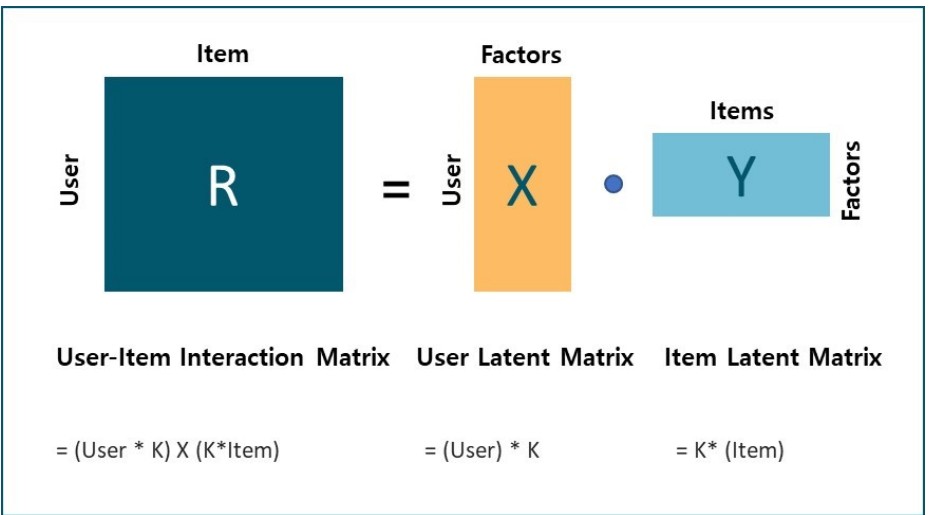

**Figure 1.** Structure of Matrix Factorization.

For this reason, when implementing a recommendation system using Implicit Feedback Data, an appropriate evaluation scale is required. The availability of an item and repeated feedback should be considered comprehensively. Availability of an item means that only one can be viewed at a time, such as in the case of two TV shows that are aired at the same time, so data is not accumulated even if you like the other show. Repeated feedback is a consideration of how users will evaluate a program differently when they watch it only once compared to when they watch it multiple times.

*2.3. Neural Collaborative Filtering*

For the existing research model [4], Neural Collaborative Filtering, referenced in this study, the User ID and Item ID of Implicit Feedback Data are input to generate each Embedding Layer and learn through the Generalized Matrix Factor (GMF) and Multi-Layer Perceptron (MLP) Layer [4,7].

A model that can express complex relationships between users and items in a more flexible way through neural net-based Neural Collaborative Filtering (NCF) while pointing out the limitations of Matrix Factorization based on the liner method in learning the relationship between users and items is needed. By proposing a Neural Matrix Factorization model that combines the linear structure of GMF and the non-linear structure of MLP, the advantages of each model are utilized to compensate for the disadvantages, and GMF and MLP use different Embedding layers.

As Figure 2 shows, the feature vectors of User and Item that correspond to the input layer are expressed in one hot encoding and are in a very sparse state [14]. A k-dimensional latent vector is created through the embedding layer, and the fully connected layer is used in the same way as the general embedding method. Each latent vector that has been embedded is learned through the Neural Collaborative Filtering Layer, and the final value is obtained by weighting the output from each model with h in the output layer.

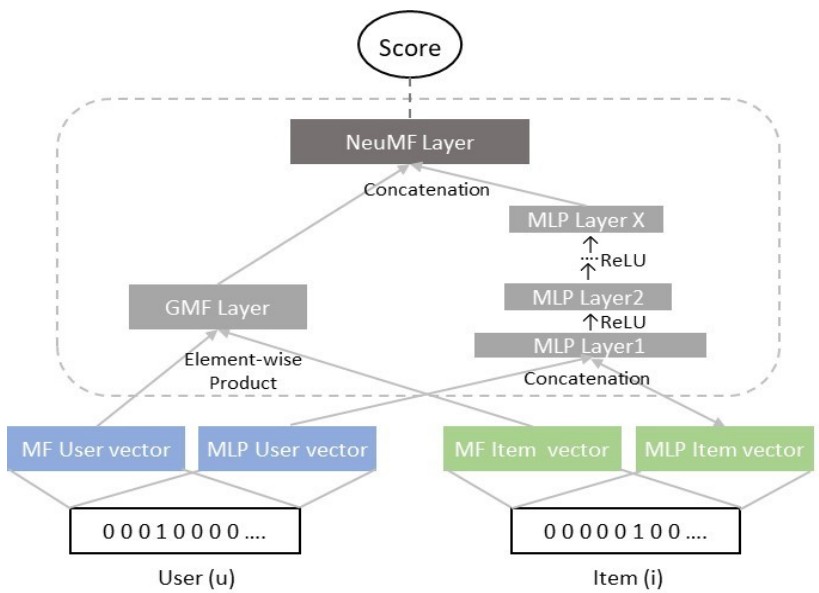

**Figure 2.** Structure of Neural Collaborative Filtering.

Since this proposes a general framework for collaborative filtering using neural networks rather than specifications of a specific model, there is a limitation in that applying this model to credit card payment data and recommending merchants does not reflect the key domain information of payment data. Therefore, this study proposes a model specialized in recommending merchants by reflecting the major domain information of credit card payment data based on the framework of the existing research model [4].

*2.4. Performance Evaluation Method*

2.4.1. Hit Rate (HR@K)

HR@K represents the number of hit users compared to the total number of users if there is a hit among Top-K recommended items for each user. It can be expressed in the following way.

$$Hit\ Rate@K = \frac{Number\ of\ Hit\ Users}{Number\ of\ Users} \tag{2}$$

2.4.2. Normalized Discounted Cumulative Gain (NDCG@K)

Cumulative Gain (CG) represents the sum of the relevance of the recommended item. Relevance is a value that indicates how related the user and the item are and can be defined by whether it is clicked or not. If the same set of items is recommended regardless of the order, the CG of the two models is the same. Discounted Cumulative Gain (DCG) is the application of the concept of order to CG. The lower the order of the recommended items, the higher the denominator value. However, if the number of items recommended to each user is different, it is difficult to compare performance accurately through DCG. Normalized Discounted Cumulative Gain (NDCG) compensates for these limitations. It can be obtained by dividing DCG by IDCG and normalizing it. IDCG is the value when the best recommendation is made among DCGs.

That is, NDCG@K is an index indicating how good the current recommendation list is compared to when it is recommended with the ideal combination and has a value between 0 and 1. The closer to 1, the higher the performance.

$$NDCG@K = \frac{DCG}{IDCG} = \sum_{i=1}^{k}\left(\frac{rel_i}{\log_2(i+1)}\right) / \sum_{i=1}^{k}\left(\frac{rel_i^{opt}}{\log_2(i+1)}\right) \tag{3}$$

## 3. Proposed Method

The model proposed in this study is based on the framework of [4]. It is based on the Neural Matrix Factorization (NMF) structure that combines Generalized Matrix Factorization (GMF) and Multi-Layer Perceptron (MLP). In order to implement a model optimized for recommending potential payment merchants using credit card payment data, the following were reviewed [4,6,7,15,16].

### 3.1. Performance Evaluation Method of Models That Reflect the Domain Information of Credit Card Payment Data

In order to recommend stores that a specific customer is likely to use in the future based on credit card payment data, it is important to understand the characteristics of the customer and the stores used by the customer. The most basic factors that can identify customer characteristics are gender and age information. This is because consumption patterns vary depending on gender, and the items consumed by age groups often vary. In addition, the industry information of a store is the most basic data that indicates the character of the store, and region information also plays a very important role in identifying customer consumption. For example, if a member store located in Busan is recommended to a customer living in Seoul, the possibility of actual consumption will decrease.

In the existing research [4], the proposed model learns using only the user ID and item ID. However, when this structure is applied to card payment data as is, there is a limitation in that the major domain information required for the recommendation of the stores mentioned above is not reflected. To supplement this, this study implemented a model optimized for merchant recommendation by adding learning data and changing the model structure accordingly.

In the same way as Figure 3 shows the structure proposed in the existing research model [4], it was learned with only the customer ID and the store ID.

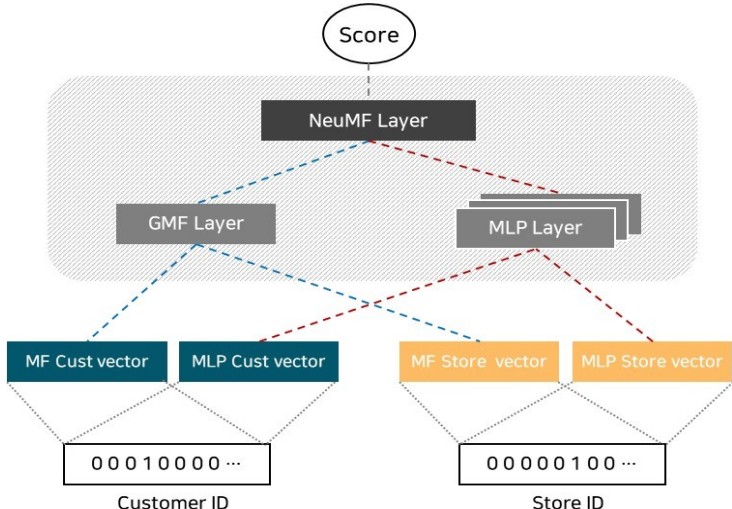

**Figure 3.** Structure of Base NMF.

As Figure 4 shows, Customer Information (gender, age) was added to the Base NMF to learn, and the Customer Latent Vector was calculated by combining Customer ID and Customer Information.

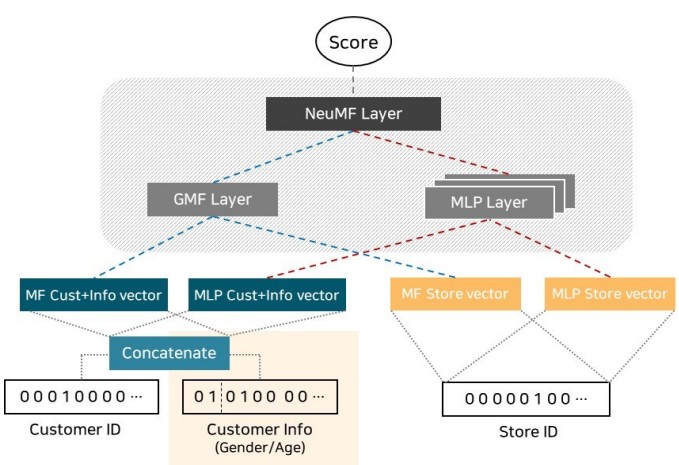

**Figure 4.** Structure of NMF with Customer Information (NMF_CI).

Store Information (industry, region) was added to the Base NMF to learn, and the Store Latent Vector was calculated by combining Store ID and Store Information as Figure 5 shows.

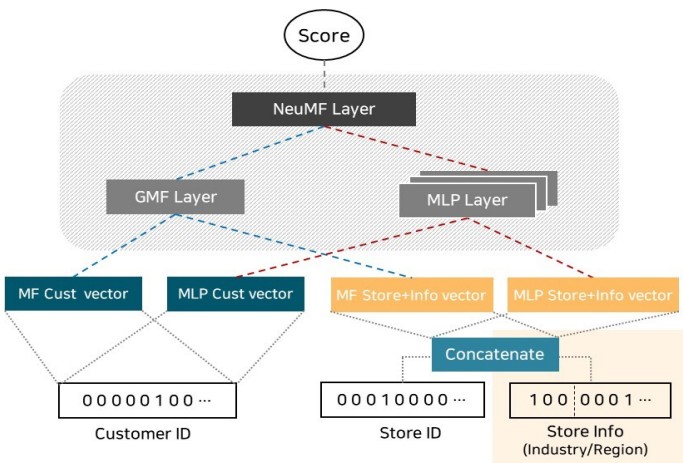

**Figure 5.** Structure of NMF with Store Information (NMF_SI).

As Figure 6 shows, both Customer Information and Store Information were added to the Base NMF to learn, Customer ID and Customer Information were combined, and Store ID and Store Information were combined to calculate each Latent Vector.

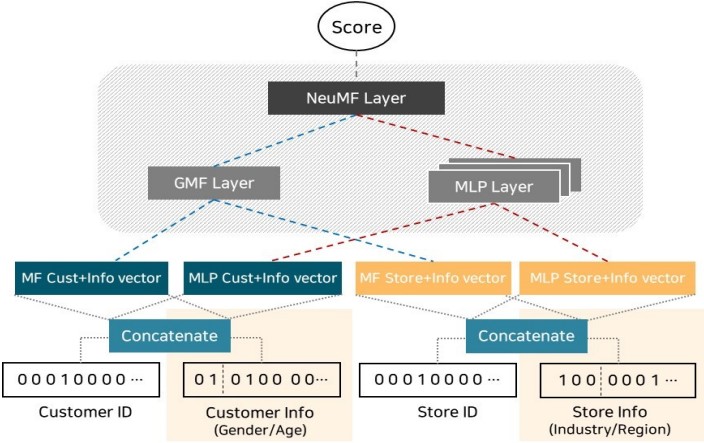

**Figure 6.** Structure of NMF with Customer and Store Information (NMF_CSI).

### 3.2. Negative Sampling

When creating a recommendation system model, past feedback information is used to learn user preferences. In this process, information on both the products preferred and non-preferred by users is needed. However, there is no clear label for user preferences in Implicit Feedback Data, so it only gives information on whether or not to use items. Therefore, when learning the model, all products used by the user were assumed to be positive examples, and some products not used by the user were selected as negative examples. In this way, the process of selecting a sample among products not used by the user is Negative Sampling. As discussed in previous related studies, using negative samples is a way to improve the performance of the model by allowing the recommendation model to distinguish between negative samples that are close to the correct answer [4,6,7].

In this study, Random Negative Sampling was used. Random Negative Sampling is a method of randomly selecting samples from all items not used. In the experiment, the model performance was compared while changing the number of negative samples.

### 3.3. Using a Pretrained Model

When training a model, initialization plays an important role in the performance of deep learning models. In the previous research [4], it was proposed to initialize NeuMF using the Pretrained Model. Each GMF and MLP were trained until convergence through random initialization, and the learned parameter values of each model were used as the initial values parameter of the NeuMF model. After inputting the learned parameters into NeuMF, only the parameter part of the output layer, where the weights of the two models are combined, was learned. Adam was used for pretraining GMF and MLP, and vanilla SGD was used for NeuMF training. Adam needed to store appropriate momentum information when updating parameters, as this approach was not suitable for optimizing NeuMF.

### 3.4. Coverage Expansion to Low-Performing Customers

In the existing research model [4], two datasets were used, MovieLens and Pinterest. In the MovieLens dataset, only users with at least 20 ratings were used, and in the Pinterest dataset, only users with more than 20 interactions were used for learning. However, when recommending merchants by using credit card payment data, recommendations should be made to customers with poor usage performance. Additionally, when marketing using the recommendation system, recommendations should also be made to customers with low usage. Thus, this study expanded the coverage to include low-performance customers. According to the number of stores used during the analysis period, customers were divided into three groups, High, Mid, and Low, and the performance of each was compared and evaluated. The top 10% of customers were classified as High, the bottom 30% of customers were classified as Low, and the remaining customers were classified as Mid. The criteria for dividing groups are as follows:

(1)   High: Customers with more than 20 stores used;
(2)   Mid: Customers with 6 to 20 stores used;
(3)   Low: Customers with less than 5 stores used.

## 4. Experiments

### 4.1. Dataset Description

The data used in this study was the payment data from one domestic credit card company, and the analysis period was from November 2021 to March 2022. There were about 8 million customers with valid credit cards and about 1.6 million valid affiliated stores. Finally, 10,000 customers and 5000 affiliated stores were used for learning through sampling.

The criteria for sampling customers and stores were as follows. First of all, in the case of customers, after dividing into three groups according to the number of stores they used, high, mid, and low, 3000, 2000, and 1000 people were sampled, respectively, and the learning data was composed of a total of 10,000 customers. In the case of valid stores, learning may not be performed properly if very small franchises are included. Therefore,

the number of customers who visited the store was limited to the top 5000 based on payment data for the last month.

Finally, in order to reflect the domain information of the credit card payment data, the customer's gender and age information and the store's industry and region information were added. For the industry information of stores, categories based on our company's standard code were used, and regional information was used in units of cities and counties.

### 4.2. Evaluation Metrics

In this study, HR@5, HR@10, HR@20, NDCG@5, NDCG@10, and NDCG@20 were used as performance indicators.

#### 4.2.1. HR@K (Hit Rate)

HR@K is an index indicating the number of hit users among all users, and the formula is as follows:

$$Hit\ Rate@K = \frac{Number\ of\ Hit\ Users}{Number\ of\ Users} \tag{4}$$

#### 4.2.2. NDCG@K (Normalized Discounted Cumulative Gain)

NDCG@K is an indicator of how good the current recommendation list is compared to the ideal combination and has a value between 0 and 1. The closer it is to 1, the better the performance.

$$NDCG@K = \frac{DCG}{IDCG} = \sum_{i=1}^{k}\left(\frac{rel_i}{\log_2(i+1)}\right) / \sum_{i=1}^{k}\left(\frac{rel_i^{opt}}{\log_2(i+1)}\right) \tag{5}$$

### 4.3. Experimental Results

For performance comparison with the basic structural model, the GMF, MLP, and NeuMF models proposed in [4] were measured by learning only the Customer ID and Store ID of the credit card payment data in the same way as [4]. The results are shown in Table 1.

**Table 1.** Results of baseline models.

|  | GMF | MLP | NeuMF |
|---|---|---|---|
| HR@5 | 0.6551 | 0.6429 | 0.6592 |
| HR@10 | 0.7655 | 0.7475 | 0.7688 |
| HR@20 | 0.8578 | 0.8579 | 0.8588 |
| NDCG@5 | 0.5104 | 0.5067 | 0.5107 |
| NDCG@10 | 0.5429 | 0.5345 | 0.5408 |
| NDCG@20 | 0.5697 | 0.5677 | 0.5697 |

#### 4.3.1. Baselines

As with the results in [4], it was confirmed that most of the performance was high in NeuMF. Therefore, based on the NeuMF model structure, the major domain information of credit card payment data was sequentially reflected.

#### 4.3.2. Models That Reflect the Domain Information of Credit Card Payment Data

The NeuMF (Base NMF) model discussed above has a structure that uses only Customer ID and Store ID as input data. In order to reflect the important domain information of payment data, the NMF_CI model with customer information added, the NMF_SI model with the store information added, and the NMF_CSI model with both customer information and store information were sequentially trained to measure the performance. The experimental results are shown in Table 2.

**Table 2.** Results of models reflecting domain information.

|  | Base NMF | NMF_CI | NMF_SI | NMF_CSI |
|---|---|---|---|---|
| HR@5 | 0.6592 | 0.6446 | 0.6709 | 0.6840 |
| HR@10 | 0.7688 | 0.7541 | 0.7823 | 0.7899 |
| HR@20 | 0.8564 | 0.8523 | 0.8768 | 0.8811 |
| NDCG@5 | 0.5107 | 0.5032 | 0.5236 | 0.5335 |
| NDCG@10 | 0.5408 | 0.5341 | 0.5600 | 0.5652 |
| NDCG@20 | 0.5659 | 0.5587 | 0.5842 | 0.5891 |

The performance of NMF_CI with customer information was slightly lower than that of Base NMF, but the performance of NMF_SI with store information was improved, and it was confirmed that the NMF_CSI model, with both customer information and store information, was the highest in all evaluation indicators. Since the number of recommendations, K, was set to 10 in the primary research on recommendation systems, we compared the results of HR@10 and NDCG@10 [17–19].

In addition, the performance of each customer group (High/Mid/Low) was as Tables 3 and 4 show.

**Table 3.** Results by group of models reflecting domain information (HR@10).

|  | Base NMF | NMF_CI | NMF_SI | NMF_CSI |
|---|---|---|---|---|
| High | 0.7832 | 0.7676 | 0.7960 | 0.8002 |
| Mid | 0.7633 | 0.7510 | 0.7820 | 0.7877 |
| Low | 0.7274 | 0.7110 | 0.7319 | 0.7565 |

**Table 4.** Results by group of models reflecting domain information (NDCG@10).

|  | Base NMF | NMF_CI | NMF_SI | NMF_CSI |
|---|---|---|---|---|
| High | 0.5519 | 0.5525 | 0.5727 | 0.5753 |
| Mid | 0.5381 | 0.5290 | 0.5607 | 0.5652 |
| Low | 0.5054 | 0.4768 | 0.5109 | 0.5279 |

When comparing HR@10 and NDCG@10 values for each customer group, the NMF_CSI model also showed the best performance. Through this, it can be confirmed that it is more effective to learn the interaction between customers and stores by learning the customer's gender/age information and the store's industry/region information together.

Performance by group was high in the order of High > Mid > Low in all models. In other words, it can be said that the more active the card user is, the more accurate the provided recommendation is. However, the results of the NMF_CSI model showed that the HR@10 value of the group Low was 0.7565, and the NDCG@10 value was 0.5279, which was not much different from the NeuMF results (HR@10: 0.7688, NDCG@10: 0.5408) in Table 1. Therefore, the merchant recommendation model proposed in this study shows that it is possible to expand and apply coverage to low-performance customers.

### 4.3.3. Model Comparison by the Number of Negative Samples

The Loss Function can be divided into the Pointwise method and the Pairwise method according to the number of items considered at once when learning the model. The Pointwise method considers one item at a time. Scores are obtained for all items and ranked by sorting. In the case of the Pairwise method, two items are considered as pairs at a time. It is a method of finding an optimized order in which the answer list and pairs match.

When using the Pairwise method, only one negative instance should be considered for one positive instance, but in this study, the sampling ratio for negative samples can be freely set because negative instances were learned together using Pointwise log loss.

Even with negative sampling, the performance of NMF_CSI was the best, and the performance was compared by changing the number of negative samples learned at once for

the NMF_CSI model. The results are shown in Table 5. Initially, the performance seemed to improve as the sampling ratio increased, but it was confirmed that if there were more than seven, the performance decreased. The optimal sampling ratio ranges from three to six.

**Table 5.** Results according to the number of negative samples.

| #Neg | HR@10 | NDCG@10 |
|------|-------|---------|
| 1 | 0.7695 | 0.5368 |
| 2 | 0.7826 | 0.5502 |
| 3 | 0.7919 | 0.5625 |
| 4 | 0.7899 | 0.5652 |
| 5 | 0.7909 | 0.5651 |
| 6 | 0.7936 | 0.5682 |
| 7 | 0.7889 | 0.5694 |
| 8 | 0.7863 | 0.5724 |
| 9 | 0.7842 | 0.5723 |
| 10 | 0.7807 | 0.5730 |

4.3.4. Comparison of Models with or without the Pretrained Model

After learning using the pretrained model of GMF and MLP, the performance was compared. In both HR@10 and NDCG@10, using the pretrained model was higher. Each result is shown in Table 6.

**Table 6.** Results according to whether Pretrained Model is used or not.

| | Without Pretrain | | With Pretrain | |
|---|---|---|---|---|
| | HR@10 | NDCG@10 | HR@10 | NDCG@10 |
| NMF | 0.7688 | 0.5408 | 0.7835 | 0.5642 |
| NMF_CI | 0.7542 | 0.5341 | 0.7712 | 0.5547 |
| NMF_SI | 0.7823 | 0.5600 | 0.8033 | 0.5762 |
| NMF_CSI | 0.7899 | 0.5652 | 0.8081 | 0.5853 |

**5. Conclusions**

As the market for credit card payments has grown, the importance of personalized recommendations has increased in the credit card industry. However, if deep learning recommendation methods based on content recommendation, collaborative filtering, or simple user-item interaction are applied as is, there is a limit to the reflection of the main domain characteristics of credit card payment data. Therefore, in this study, we reconstructed learning data by adding customer gender and age information, merchant industry, and region information to implement a model optimized for recommending merchants with a high possibility of future payment by using credit card payment data. We have also expanded our coverage so that these results can be applied to underperforming customers. To verify the excellence of the NM_CSI model proposed in this study, payment data from a credit card company with more than 8 million customers collected in Korea was used. As a result of c experiments comparing the basic NMF model and the proposed NM_CSI model, performance improved by 3% based on HR@10 and 5% when based on NDCG@10. These results are expected to be used in various ways for research on affiliate store recommendations using credit card payment data.

**Author Contributions:** Conceptualization, S.Y. and J.K.; methodology, S.Y.; software, S.Y.; validation, S.Y. and J.K.; formal analysis, S.Y. and J.K.; investigation, S.Y. and J.K.; resources, S.Y.; data curation, S.Y.; writing—original draft preparation, S.Y.; writing—review and editing, J.K.; visualization, S.Y. and J.K.; supervision, J.K.; project administration, S.Y. and J.K.; funding acquisition, J.K. All authors have read and agreed to the published version of the manuscript.

**Funding:** This research was supported by the MSIT (Ministry of Science and ICT), Korea, under the ICT Creative Consilience Program(IITP-2023-2020-0-01821) supervised by the IITP(Institute for Information & communications Technology Planning & Evaluation)".

**Data Availability Statement:** Data is unavailable due to privacy or ethical restrictions.

**Conflicts of Interest:** The authors declare no conflict of interest.

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
