# Peer review of "Merchant Recommender System Using Credit Card Payment Data"

_electronics, doi:10.3390/electronics12040811_

Round 1

Reviewer 1 Report

Authors describe a recommendation system for credit card payment analysing transactions in Korea.

Introduction and related works include sufficient descriptions.

The workflow is sufficient described. The English is fine.

The references are insufficient. The listed items are not linked in the text and a lot of number or names are not referenced. An incomplete list includes: source of Korean numbers (rows 35, 39, 40, 42, 290, 291), neural matrix factorization, explicit feedback data,  GMF, MLP, random negative sampling, pretrained model, Adam, Vanilla SGD, MovieLens, Pinterest, Pointwise, etc.

Authors must check document and add references.

Figure 3, 4 and 5 should be removed and their texts included in current paragraph 3.1.4.

I suppose "4.Conclusions" is  Results.

Authors should clearly explain what are 5, 10 and 20 in HR@K and NDCG@K.

It is not clear because authors focus on HR@10 and NDCG@10  while HR@20 and NDCG@20 have better values in Table2.

It should be interesting to have and a discuss values for HR@5, NDCG@5, HR@20 and NDCG@20 on NMF models like values for HR@10 and NDCG@10 in tables 3 and 4.

A related works section should be added or included in "Conclusions".

Why the last sentence "Finally, 10,000 customers."?

Author Response

Thank you for your comments. I revised my paper according to your comments. And the followings are the responses point-by-point.

The references are insufficient. The listed items are not linked in the text and a lot of number or names are not referenced. An incomplete list includes: source of Korean numbers (rows 35, 39, 40, 42, 290, 291), neural matrix factorization, explicit feedback data,  GMF, MLP, random negative sampling, pretrained model, Adam, Vanilla SGD, MovieLens, Pinterest, Pointwise, etc.
Authors must check document and add references.
(Ans.) I checked and added references in the whole manuscript.

Figure 3, 4 and 5 should be removed and their texts included in current paragraph 3.1.4.
(Ans.) According to your comment, I have combined figures 3, 4, and 5 into one paragraph.

I suppose "4.Conclusions" is  Results.
(Ans.) Thank you for the comment. I fixed it.

Authors should clearly explain what are 5, 10 and 20 in HR@K and NDCG@K.
It is not clear because authors focus on HR@10 and NDCG@10  while HR@20 and NDCG@20 have better values in Table2.

It should be interesting to have and a discuss values for HR@5, NDCG@5, HR@20 and NDCG@20 on NMF models like values for HR@10 and NDCG@10 in tables 3 and 4.
(Ans.) I added the follow sentence under the table 2. "Since the number of recommendations, K, was set to 10 in the main research on recommender systems, we will compare the results of HR@10 and NDCG@10 [17][18]."

A related works section should be added or included in "Conclusions".
(Ans.) I included relaed works in conclusions.

Why the last sentence "Finally, 10,000 customers."?
(Ans.) I revised conclusion part including the last sentaence.

Thank you, Best regards.

Reviewer 2 Report

A recommendation system is proposed which is based on the credit card payment information data. The presentation is clear with fine analysis. I recommend the acceptance of this paper after revision.

1. More introductions about the benefit from the model comparison of the number of negative samples would be better.

2. One paper related to the recommendation system based on the collaborative information, the authors may introduce this paper.

    Ja-Hwung Su, Wei-Yi Chang, and Vincent S. Tseng, “Integrated Mining of Social and Collaborative Information for Music Recommendation,” Data Science and Pattern Recognition, vol. 1(1), pp. 13-30, 2017

3. The classification of proposed method is based on the MLP, the authors may introduce more about the concep of the MLP.

Author Response

Thank you for your comments. According your comments, I revised and updated the manuscript. And followings are answers of your comments.

A recommendation system is proposed which is based on the credit card payment information data. The presentation is clear with fine analysis. I recommend the acceptance of this paper after revision.

1. More introductions about the benefit from the model comparison of the number of negative samples would be better.
(Ans.) I added the following sentence at the line 244-247.
"As discussed in previous related studies, using negative samples is a way to improve the performance of the model by allowing the recommendation model to distinguish between negative samples that are close to the actual correct answer and the actual correct answer[4][6][7]."

2. One paper related to the recommendation system based on the collaborative information, the authors may introduce this paper.

    Ja-Hwung Su, Wei-Yi Chang, and Vincent S. Tseng, “Integrated Mining of Social and Collaborative Information for Music Recommendation,” Data Science and Pattern Recognition, vol. 1(1), pp. 13-30, 2017
(Ans.) Thank you for your suggestion. I added the reference as [19] in the manuscript.

3. The classification of proposed method is based on the MLP, the authors may introduce more about the concep of the MLP.
(Ans.) Thank you for you comment. According to your comment, I replaced the concept to MLP with references at the line 133.

Thank you. Best regards.

Round 2

Reviewer 1 Report

Just one little thing: probably one "." is missing in row 242.